# Molecular Hydrogen: From Molecular Effects to Stem Cells Management and Tissue Regeneration

**DOI:** 10.3390/antiox12030636

**Published:** 2023-03-03

**Authors:** Mikhail Yu. Artamonov, Andrew K. Martusevich, Felix A. Pyatakovich, Inessa A. Minenko, Sergei V. Dlin, Tyler W. LeBaron

**Affiliations:** 1Laboratory of Translational Free Radical Biomedicine, Sechenov University, 119991 Moscow, Russia; 2MJA Research and Development, Inc., East Stroudsburg, PA 18301, USA; 3Laboratory of Medical Biophysics, Privolzhsky Research Medical University, 603005 Nizhny Novgorod, Russia; 4Department of Kinesiology and Outdoor Recreation, Southern Utah University, Cedar City, UT 84720, USA; 5Molecular Hydrogen Institute, Enoch, UT 84721, USA

**Keywords:** molecular hydrogen, oxidative stress, mechanisms, mesenchymal stem cells, regeneration

## Abstract

It is known that molecular hydrogen is a relatively stable, ubiquitous gas that is a minor component of the atmosphere. At the same time, in recent decades molecular hydrogen has been shown to have diverse biological effects. By the end of 2022, more than 2000 articles have been published in the field of hydrogen medicine, many of which are original studies. Despite the existence of several review articles on the biology of molecular hydrogen, many aspects of the research direction remain unsystematic. Therefore, the purpose of this review was to systematize ideas about the nature, characteristics, and mechanisms of the influence of molecular hydrogen on various types of cells, including stem cells. The historical aspects of the discovery of the biological activity of molecular hydrogen are presented. The ways of administering molecular hydrogen into the body are described. The molecular, cellular, tissue, and systemic effects of hydrogen are also reviewed. Specifically, the effect of hydrogen on various types of cells, including stem cells, is addressed. The existing literature indicates that the molecular and cellular effects of hydrogen qualify it to be a potentially effective agent in regenerative medicine.

## 1. Introduction

In recent decades, medical gases have deservedly attracted much attention from specialists in the field of biomedicine. At the same time, the spectrum of gases for which biological effects have been discovered and a beneficial effects have been observed is quite wide and continues to increase [1,2,3]. Initially, the concept of “medical gases” extended only to oxygen and hyperbaric O_2_ procedures, followed by the introduction of the inhalation of anesthetics and ozone into medical practice. At the same time, among these gases, only ozone had specific bioregulatory effects, while other factors either contributed to the elimination of hypoxia, or had a systemic anesthetic effect, being considered as a pharmacological (synthetic) agent.

The prospect of using endogenously formed low-molecular-weight gaseous compounds as regulators began to arise only after the establishment of the biological activity of nitrogen monoxide (i.e., nitric oxide; NO^•^), for which the Nobel Prize in Medicine and Physiology was awarded [4]. Later, the ability of other substances—previously considered exclusively as toxic—to act as endogenous gaso-transmitters was discovered. These substances formed a triumvirate, which, in addition to nitric oxide, included hydrogen sulfide and carbon monoxide [5].

In parallel, ideas were developed about the biological properties of inert gases (helium, argon, xenon, etc.) which are chemically inactive under standard conditions, but show a modulating effect on cells and tissues [6,7,8,9]. These ideas served as the basis for the creation of innovative therapeutic technologies, the prospects of which continue to be studied and expanded upon.

Recently, molecular hydrogen, which has several unique characteristics, has occupied a special place in medical gas therapy. This gas, which has no specific color or smell, is created from the lightest chemical element, is ubiquitous, and, due to its size and minimal molecular weight, can penetrate through any biological barrier [10,11,12]. Many supplements and antioxidants require specific transporters to enter the cells and realize their effect, whereas H_2_ does not need them due it is smaller molecular weight/size (Figure 1). The high bioavailability of H_2_ satisfies the first requirement of any pharmacological agent to have a biological effect. However, there are additional properties that are needed to explain the breadth of the biomodulating and, consequently, therapeutic effect of this gas.

The body of knowledge in the field of H_2_ biomedicine is being actively updated, including about 100 randomized controlled trials and more than 2000 articles that have been published by the end of 2022 (Figure 2). Various aspects of the issue are considered in detail, but the focus of research is largely shifted towards the cardiac [11,13], neurological [14,15], and radioprotective effects [16,17] of hydrogen. Several studies assessed the possibility and expediency of using hydrogen in oncological diseases [18,19,20]. On the other hand, cellular effects, which are fundamentally significant for regenerative medicine, are revealed only indirectly. In this regard, the purpose of this review is to systematize ideas about the nature, features, and mechanisms of the influence of H_2_ on cells, specifically stem cells.

## 2. A Brief Outline of the History of the Discovery of Molecular Hydrogen as a Biological Agent and the Formation of Hydrogen Biomedicine

Hydrogen in free form was first experimentally obtained in 1671 by Robert Boyle [21] and identified as an independent chemical element in 1766 by Henry Cavendish [22]. Interestingly, by the end of the eighteenth century, Thomas Beddoes had completed the first documented attempt at the medical use of hydrogen for the treatment of patients with tuberculosis [23,24]. Until around 1969, the possibility of endogenous production of H_2_ in the human body from intestinal bacteria was not known [25,26].

One of the earliest direct uses of hydrogen was in deep sea diving. For example, Lanphier (1972) used Hydroliox (a mixture of hydrogen, helium, and oxygen) to prevent the development of decompression syndrome and nitrogen anesthesia in divers working at great depths [27]. One of the most famous early publications in this field belongs to Doyle et al. (1975), who showed a marked regression of squamous skin carcinoma in mice under the influence of H_2_ supplied under increased pressure (8 ATM) [26]. They used a gas mixture containing 97.5% hydrogen and 2.5% oxygen. Since that time, isolated reports have appeared on the effectiveness of the use of H_2_ for other applications [28,29]. However, an article in 2007 by Ohsawa et al. was the trigger for an rapid increase in the interest of specialists into the biological and medical effects of hydrogen [30]. The article presents the results of a successful treatment using inhalation of H_2_ to prevent the damage induced from ischemia–reperfusion after ischemic stroke in a rat model [30]. The researchers suggested that the main mechanism for achieving this clinical effect is the relief of oxidative stress induced by these pathological conditions.

Most recently, hydrogen therapy was included in the protocol for the management of patients with COVID-19 in China (Chinese Clinical Guidelines (7th edition) for COVID-19 Pneumonia Diagnosis and Treatment, issued by China National Health Commission). For this purpose, based on extensive experimental and clinical studies, it appears that inhalation of a mixture of 66.6% H_2_ and 33.3% oxygen significantly reduces the rate of deterioration of respiratory lung function in a new coronavirus infection, as well as the development of emphysema and inflammatory reactions in lung tissue in various acute and chronic diseases [31].

## 3. Routes of Introducing Molecular Hydrogen into the Body

Currently, the range of routes of introducing H_2_ into the body is extremely wide (Figure 2). First, it is important to emphasize that these methods differ not only in the convenience of application for a specific pathology (e.g., in the case of the treatment of dermatological diseases, hydrogen baths may be the preferred option), but also in the pharmacokinetics of the molecule, which alters its pharmacological activity [32].

Historically, the first way of introducing hydrogen is the use of hyperbaric chambers with an atmosphere rich in hydrogen gas [26]. Despite the encouraging results obtained in the experiments by Dole et al. (1975), work using such a technology was not continued, which may be due to the difficulties of its practical implementation [26].

The most common options for molecular hydrogen therapy are the inhalation of H_2_-containing gas mixtures of various compositions, the use of hydrogen-saturated water, and the infusion/injection of a sodium chloride solution saturated with H_2_ (Figure 2) [12]. Each of these pathways has its own characteristics, advantages, and disadvantages, as well as possible different molecular mechanisms of action.

Most studies aimed at evaluating the effectiveness of the use of H_2_ have used hydrogen-saturated solutions (i.e., hydrogen water, hydrogen saline, etc.). However, studies on the inhalation of hydrogen, especially in clinical use, are increasing [11,17,30,31]. Inhalation of H_2_ is a fairly simple way of exposure for both laboratory animals and humans. Indeed, this was the method employed by Ohsawa et al. using the rat model of ischemia–reperfusion [30]. In addition, an important advantage of the technology is the possibility of the strict dosing of hydrogen by regulating the exposure time and concentration of H_2_ in the gas mixture [32,33]. On the other hand, molecular hydrogen is a combustible and explosive gas in the case of its reaction with oxygen. It is assumed that the risk of such a negative effect is quite high when the concentration of H_2_ in the gas mixture is higher than 4% [17,32,33]. Nevertheless, in some cases, gas mixtures with a high hydrogen content are used, but with special safety requirements. For example, 66.67% H_2_ and 33.33% O_2_ is used in the treatment of patients with COVID-19, which is the same hydrogen therapy protocol that has been introduced in China [34]. The effectiveness of H_2_ inhalation in chronic obstructive pulmonary disease [35,36] and severe bronchial asthma [37] has also been reported. The expediency of such an approach is associated with the variable dose dependence of the antioxidant and anti-inflammatory properties of H_2_ [32,38].

Considering the physicochemical characteristics and the extremely small size and molecular weight of H_2_, hydrogen inhalation has ample opportunities for systemic action. Since it easily diffuses through the walls of the alveoli, hydrogen diffuses into the blood plasma and is transported to various organs and tissues. Experimental studies by Cole et al. (2021) have shown that in healthy animals, inhalation of a gas mixture containing 2.4% hydrogen continuously for 72 h does not cause any changes in physiological parameters [39]. At the same time, sanogenetic effects on various organs and tissues have been demonstrated in numerous studies [31,32,33,34,35,36,37].

The most convenient way of introducing molecular hydrogen in clinical practice is drinking water saturated with molecular hydrogen; this option eliminates the explosion and fire hazard of the therapy and ensures its portability, opening up the possibility for the widespread use of H_2_-containing water. However, this path also has disadvantages associated with low gas solubility [32]. It is known that the saturation of dissolved hydrogen is 0.78 mM (1.57 mg/L) at normal atmospheric pressure and room temperature [40]. This circumstance may be significant since it does not always allow achieving the necessary dose of the molecule to ensure a full clinical effect. In addition, when using this route, it should be considered that the prepared hydrogen water should be applied immediately, since it has a very short period of maintaining the concentration of H_2_. Moreover, upon ingestion of hydrogen water, a significant amount (>90%) is lost via normal expiration [41]. This at least indicates the high uptake of H_2_ to pass through the gastrointestinal tract and into the venous system where it reaches the lungs and is exhaled. At the same time, the distribution of hydrogen in various tissues and organs after drinking hydrogen water is not the same. In particular, the penetration of H_2_ into brain cells when using the route of administration under consideration is minimal [42], which may be of fundamental importance for determining indications for its clinical use.

These above reasons necessitated the search for alternative ways of delivering H_2_ to tissues, including the creation of nanocomposites with a delayed release of gas [43]. It was assumed that these nanocomposites would be included in the oral tablet form, which would ensure maximum compliance for patients. Such targeted H_2_ therapy can be carried out using hybrid palladium nanocrystals. This approach was tested in experimental conditions in a model of oncopathology, which allowed not only the confirmation of its anticarcinogenic activity, but also its protection of unchanged cells against hyperthermia [44], as well as the manifestation of oxidative stress and damage induced by ischemia and reperfusion [43,45]. The possibility of using various elements as a basis for hydrogen-releasing nanocomposites (in particular, silicon particles [43]) should be emphasized.

A technology fundamentally similar to the creation of nanocrystals as carriers of H_2_ is the use of microbubble systems (Figure 2). In recent years, it has been suggested that a specific delivery of hydrogen using microbubbles provides maximum bioavailability and a minimization of “transport losses” of the molecule [46]. An important advantage of this pathway is the possibility of introducing significantly higher amounts of gas compared to the intake of water that has been saturated with H_2_. The effectiveness of the method was demonstrated in a model of ischemic myocardial injury in rats [46].

The third main route of introducing H_2_ into the body is the use of injections and infusions of an H_2_-saturated isotonic saline solution [17,32,33]. The specified path also has advantages and disadvantages. For one, this path allows the dosing of the injected amount of hydrogen with high accuracy, the use of different concentrations, an increase in the bioavailability of the agent to the target organ, and, if necessary, a carrying out of the topical effects on strictly defined areas of tissues (e.g., surface localization or areas of catheterization and injection). At the same time, injections of hydrogen solutions pose a certain risk of invasiveness and, consequently, infection, while also requiring the involvement of experienced medical personnel for manipulation. The administration of these solutions is mainly performed intravenously (in patients) or intraperitoneally (in experimental studies using laboratory animals) [17]. In our opinion, the clinical potential of this route of administration is not fully developed, as evidenced the rich experience of intravenous ozone therapy, also based on the effects of a medical gas [47,48,49].

Currently, the literature describes alternative routes of using molecular hydrogen, which use both its systemic and local action (Figure 2). In particular, an example of the effective administration of H_2_ in dermatology and cosmetology is the use of hydrogen baths for psoriasis [50] and liposuction [51]. A variant of H_2_ therapy close to this technology is the use of eye drops saturated with the gas in question for the treatment of ischemic lesions of the iris of the eye, as well as the suppression of apoptosis [52]. Stimulation of the endogenous synthesis of H_2_ by symbiotic microflora is of particular interest. It has been shown that oral administration of lactulose increases the synthesis of H_2_ by bacteria of the gastrointestinal tract [53]. Moreover, a hydrogen breathing test has been proposed to determine the state of the intestinal microflora by its ability to generate molecular hydrogen [54]. It is important to emphasize that hydrogen is synthesized by microorganisms together with other gases (for example, methane) [55]. At the same time, the amount of hydrogen produced is the result of the balance of activity of the H_2_-producing (hydrogenogenic) and H_2_-utilizing (hydrogenotrophic) intestinal microbes [56].

A few more exclusive H_2_ pathways should also be noted, including probe feeding with the inclusion of a solution saturated with H_2_ [57], introduction during hemodialysis [58], local treatment of the skin surface [59], the addition of hydrogen to the preservation medium for transplanted organs to prevent cold damage [60], and the washing of various body cavities.

As mentioned previously, different routes of administration differ not only in their proximity to the point of exposure, but also in their pharmacokinetics [32]. In particular, it was found that the concentration of hydrogen in the blood increases rapidly after inhalation, but 3 min after cessation, it decreases to 1/40 of the peak value [61]. At the same time, the amount of hydrogen in arterial blood during inhalation always exceeds that in venous blood, which may indicate the diffusion of the gas into the tissues [61]. It is also shown that the peak values during inhalation of H_2_ and the intake of hydrogen water are achieved at the same time (e.g., 10–30 min depending on concentration/dose). However, the duration that H_2_ remains in the body before returning to baseline lasts longer than 30 min [62]. It should be noted that the effect on molecular cascades (e.g., on the expression of NF-KB and other regulatory proteins in liver tissue) is more pronounced for hydrogen water, and the combination of its intake with H_2_ inhalation enhances the effect [62]. Interestingly, the tissue concentration of H_2_ was significantly higher and persisted for a longer time after inhalation compared to drinking hydrogen water [42]. Naturally, with intravenous administration of isotonic solutions saturated with hydrogen, the peak breath concentration of the molecule is reached as quickly as possible, within 1 min [10].

Thus, at present, there is a wide range of routes of introducing molecular hydrogen into the body, differing not only in the topical and physico–chemical parameters, but also in the pharmacokinetics of the action of the molecule.

## 4. Characteristics of Molecular and Cellular Aspects of the Biological Action of H_2_

The combination of routes of introducing H_2_ into the body indicated in the previous section is necessary for the realization of numerous biological and beneficial effects of this gaseous molecule. These include antioxidant, anti-apoptotic, anti-inflammatory activity, the regulation of gene expression, etc. However, diverse molecular mechanisms are primarily involved that mediate the unique effects of molecular hydrogen (Figure 3).

### 4.1. Antioxidant Effects

Historically, the first and most significant molecular cascade activated by H_2_ is its effect on free radical processes in biological fluids (primarily in blood) and tissues [30]. Oxidant homeostasis is an interaction and balance of free radical generation and utilization by enzyme and non-enzyme antioxidant systems [63,64]. The spectrum of the main molecules and ions involved in free radical reactions includes a set of oxidative oxygen (e.g., hydroxyl radical, superoxide radical, ozone, singlet oxygen, hydrogen peroxide, etc.), nitrogen (e.g., nitric oxide, nitrosonium, peroxynitrite), halogen (e.g., hypochlorite), and lipid (e.g., lipooxyl radical) species [63,64,65]. Under physiological conditions, these bio-oxidants participate in various extracellular and intracellular processes, including signaling cascades, in which they act as primary or secondary messengers [63]. The majority of the reactive oxygen species inside a cell is generated in the mitochondrial electron transport chain, primarily by complexes 1 and 3 [66]. In addition, free radicals are formed with the participation of NADPH-oxidase, NO-synthases, xanthine oxidase, cytochrome p450, aldehyde dehydrogenase, hemoproteins, etc. [67]. Under the influence of adverse factors of various nature (physical, chemical, biological, etc.) or the presence of systemic or local pathology, the imbalance between the excessive generation of these oxidants and the activity of their detoxification by enzymatic and non–enzymatic antioxidant systems form a special pathological condition—oxidative stress [68,69,70,71]. This condition is one of the most universal and serves as a component of the pathogenesis of a wide range of diseases by inducing damage to cells [70,72]. Therefore, the relief of oxidative stress should be considered as a significant goal of complex treatment.

Currently, many studies have assumed that the effect of H_2_ on the state of oxidative metabolism consists of the direct capture of free radicals and a regulatory effect on antioxidant systems. The first component of the antioxidant effect of the gas in question is realized by the direct interaction of hydrogen with toxic oxidants, primarily hydroxyl (•OH) and peroxynitrite (ONOO^−^) [30]. This is of fundamental importance, since it is these oxidant molecules that have the maximum oxidative activity, which is shown using a specific fluorescent probe (2′,7′-dichlorodihydrofluorescin; DCF) (Figure 4) [73]. It should be emphasized that H_2_ is a new type of antioxidant, since it selectively disposes of ^•^OH and ONOO^−^ without rendering a significant effect on the level of hydrogen peroxide and the superoxide radical, and thus does not violate the mechanisms of cellular signaling [32,74]. It is important to emphasize that the neutralization of radicals can occur in the extracellular space (including in biological fluids), as well as in any cell compartments, including plasma and mitochondrial membranes, due to the unique small size of the H_2_ molecule, which gives it a high permeability through any biological barriers [74,75].

Hydrogen can also have an antioxidant effect by stimulating the corresponding antioxidant enzymes. In 2001, Gharib et al. demonstrated that H_2_ inhalation enhanced the catalytic properties of superoxide dismutase [76]. This effect was due to the induction of the Nrf2 pathway [77]. This factor directly increases the expression of several antioxidant enzymes and proteasomes, as well as hemoxygenase-1 [78,79]. The stimulating effect of H_2_ on catalase and myeloperoxidase activity was also noted [80]. In addition, the ability of H_2_ to inhibit the expression of genes associated with the production of peroxynitrite has been established [81].

An additional mechanism for the realization of the antioxidant effect of hydrogen is the effect on signal-regulating kinase-1 (ASK1) and the regulatory pathway of mitogen-activated protein kinase (p38-MAPK), which mediates the antioxidant activity of the compound and its anti-apoptotic properties [33]. Stimulation of these cascades leads to the inhibition of NADPH oxidase activity and, consequently, a decrease in the rate of free radical generation. The combination of these effects makes it possible to characterize H_2_ as a universal broad-spectrum antioxidant.

### 4.2. Anti-Inflammatory Effects

The anti-inflammatory effects of H_2_ are closely associated with its antioxidant and anti-inflammatory effects and involve similar or identical mechanisms in their implementation. For example, as discussed in the previous section, molecular hydrogen influences ASK1- and p38 MAPK-dependent regulatory pathways, which are also involved in inflammation. Excessive generation of bioradicals can stimulate an inflammatory response due to activation of NF-kB, the p53 gene, hypoxia-inducible factor-1α, matrix metalloproteinases, etc. [82,83].

It was revealed that at an early stage of the inflammatory reaction, the use of H_2_ reduces the degree of infiltration of neutrophils and macrophages by inhibiting the expression of cell adhesion molecules (ICAM-1) and several chemokines (including MIP-1a and MIP-2) [84]. In addition, the agent in question reduces the concentration of proinflammatory cytokines (IL-1β, TNF-α, IL-6, IFN-γ, etc.) and several colony-stimulating factors (GMCSF, G-CSF), as well as factor HMGB-1 [85]. Special attention should be paid to the data that hydrogen-saturated water inhibits the main regulatory inflammatory cascade triggered by NF-kB [86]. This factor, induced by IL-1β and TNF-α, as well as by reactive oxygen species, is an important H_2_ mediator by the following mechanisms:-inhibition of the activity of NF-kB itself and its associated cytokines [87];-blockade of factor translocation from the cytoplasm to nucleus [88,89];-an increase in the level of IkB, which hinders the binding of NF-kB to DNA [88,89,90].

Another potential mechanism of the anti-inflammatory action of H_2_ is the activation of IL-10, which directly suppresses the intensity of the inflammatory response at the local level (e.g., in wound tissues) or at the whole-body level [88,89]. Additionally, the positive effect of hydrogen on the balance of subtypes of T-helper cells and their ratios relative to T-regulatory lymphocytes and mast cells was reported [88,89,90]. These effects may contribute the increase in the concentration of IL-10 following H_2_ administration. The combined molecular response of these components due to the action of H_2_ mediates not only the anti-inflammatory, but also the immunomodulatory effect of the medical gas.

### 4.3. Anti-Apoptotic Effects

It is known that apoptosis is a classic process of programmed cell death, which allows destroying the target cell without triggering an inflammatory reaction [88]. Apoptosis, depending on its features, can acts in various physiological and pathological processes, and therefore its regulation is fundamentally important for the functioning of the body [90,91]. Modern ideas about the molecular mechanisms of apoptosis include sequential activation of cysteine proteases called effector caspases [92], endonucleases, and aggregates of Bcl family proteins [91,93].

It is shown that H_2_ can have multifactorial effects on the process of autophagy [38]. In general, this gas inhibits apoptosis by affecting the signaling pathways regulating apoptosis and its associated proteins, including phosphatidylinositol-3-kinase (PI3K), protein kinase B (Akt), and 3ß-kinase glycogen synthase (GSK3ß) [94]. The use of hydrogen negatively affects the degree of activation of the cascades, including ASK1/JNK [95], ERK 1/2, and MEK 1/2, and inhibits of the activity of caspases 3, 8, and 9, as well as the Bcl/Bax system [95,96]. In addition, the anti-apoptotic effect of H_2_ is closely associated with the two effects described above, since reducing the severity of inflammatory reactions and relieving oxidative stress reduces the need for cell removal by apoptosis [97]. A similar effect is exerted by influences that contribute to maintaining the integrity and functional activity of mitochondria [98]. The regulation of autophagy serves to reduce the intensity of apoptosis, forming an important balance between these processes.

It should be noted that in some cases, the use of H_2_ can also have a pro-apoptotic effect. In particular, in some types of cancer, the medical gas stimulates early and late apoptosis, which makes it possible to effectively remove tumor cells from the body, reduce the level of proliferation in the tumor tissue, and increase the rate of its destruction [99]. At the same time, the mechanisms underlying such a paradoxical effect have not been practically studied. Thus, it is possible to state the specific directional nature (promotion or inhibition) of the influence of H_2_ on the processes of apoptosis.

### 4.4. Regulation of Pyroptosis

Pyroptosis is a relatively new form of programmed cell death, fundamentally different from apoptosis in that it induces an inflammatory reaction. Inflammation in this case is provoked by the activation of pattern-recognizing receptors in special structures called inflammasomes [100]. It is believed that this mechanism has a protective quality, but excessive stimulation of this process contributes to the development or progression of pathology. The factors contributing to the induction of pyroptosis include reactive oxygen species, casapase-1, and the inflammasomes [101]. According to the data presented above, molecular hydrogen has an anti-inflammatory and antioxidant effect, which makes it possible to block all these factors, preventing excessive activation of pyroptosis. In addition, hydrogen can inhibit pyroptosis-initiated inflammatory diseases by reducing free radicals as substrates of oxidative stress and NLRP3 depression, which allows the preservation of tissue function and its micro- and macrostructure [102].

### 4.5. Modulation of Autophagy

Autophagy, also called partial macro-autophagy, is a catabolic process that supports cellular homeostasis and is implemented with the participation of lysosomes and the ubiquitin–proteasome system [103]. This process, which initially has exclusively physiological significance, under certain conditions (in particular, under the action of stressors of abnormal intensity and/or duration) acquires a disadaptive character [104]. It has been shown that H_2_ can have a dualistic effect on the autophagy process. On the one hand, hydrogen can stimulate a specific nucleotide-bearing domain and inhibit NLRP3 in macrophages and limit inflammatory reactions [88,105]. This makes it possible to achieve a protective effect by stimulating autophagy and providing an “intracellular renewal” procedure.

On the other hand, in some pathological conditions, autophagy acquires the character of being impaired or excessive. Under these conditions, it is advisable not to stimulate, but to inhibit this process [33]. In particular, in acute pulmonary pathology caused by lipopolysaccharide of pathogenic bacteria, the use of H_2_ limits excessive autophagy [33]. Similarly, in brain injuries, hydrogen therapy contributes to the inhibition of autophagy, which increases the survival of endothelial cells in the microcirculatory bed of brain tissue [106].

### 4.6. Cellular and Tissue Effects of Molecular Hydrogen

The numerous molecular effects of molecular hydrogen naturally cause the formation of shifts in the functioning of cells and tissues, and the nature of the modulating action of H_2_ directly depends on the path of the compound entering the body (Figure 5). Thus, when using H_2_ in the form of inhalation or through the intake of H_2_-saturated water, tissue barriers must be overcome for dissemination throughout the body (hemato–alveolar or enteral). In these locations, it is assumed that the local action of the compound will be realized while hydrogen will penetrate the alveolocytes/enterocytes, inducing a combination of the molecular effects shown in the diagram. The most significant among these effects are the antioxidant effect, post-translational modifications, shifts in the level of phosphorylation of various proteins, and the modulation of the structural and functional status of mitochondria. In addition, H_2_ entering the nucleus is possible, and regulation of gene expression, as well as epigenetic control—perhaps via the prevention of oxidative modification of DNA and histone proteins—can occur.

After overcoming tissue barriers at the site of primary contact, as well as when using injection of hydrogen solutions, H_2_ enters the blood, where it also has several significant effects. In this biological fluid, the cornerstone action is the modulation of oxidative metabolism with the capture of toxic radicals and the rejuvenation of endogenous antioxidants. However, alternative effects should also be considered, including the effect on blood clotting factor and surface receptor structures. In principle, an anti-atherogenic effect is also possible due to the protection of endotheliocytes with respect to oxidative damage.

Further transport of H_2_ to various organs and tissues ensures the development of systemic effects, which include:-organoprotective effects;-minimization of the consequences of ischemic–reperfusion lesions;-limitation of systemic inflammatory responses;-antitumor effects;-anti-aging effects;-increasing the body’s resistance to stressors of various nature;-improved exercise tolerance.

In addition, there is evidence for the modulating effect of molecular hydrogen on various types of metabolism (in particular, lipid, protein, and carbohydrate metabolism) [107]. Perhaps a combination of molecular/cellular, and systemic effects presupposes a regulatory role of H_2_ in the management of the state of various cells, including stem cells.

## 5. The Effect of Molecular Hydrogen on Various Cell Pools and Regeneration Processes

As already mentioned, the multifactorial biological activity of H_2_ creates prerequisites for its modulating effect on the processes of cell formation, starting with early precursors—stem cells. This hypothesis is confirmed in the available experimental data obtained in vitro and in vivo and testifying to the positive effect of H_2_ on all stages of the formation of differentiated cells (Figure 6).

In studies performed on the model of aplastic anemia, it was shown that the use of H_2_ influences mesenchymal stem cells, as it contributes to an increase in the number of colony-forming units [85,108]. In addition, the introduction of hydrogen positively modulates the state and functional activity of mitochondria in various cancer cells [109]. In particular, this is ensured by the activation of genes responsible for the synthesis of components of complex I of the electron transport chain [109]. The second factor is the fact that the benefits of H_2_ to the mitochondria include an increase in the mass of this organelle, the concentration of the superoxide radical within it, and an increased membrane potential [109]. These effects together provide an increase in cellular proliferation, at least in some types of cancer cells.

Special attention should be paid to the effect of H_2_ on mtUPR as a recently identified mediator involved in mitochondrial function [110] and cell survival in adverse conditions [111]. This effect may be associated with the stimulation of HSP production [112]. The stimulating effect of hydrogen on mtUPR is manifested as a change in the level of phosphorylation of eIF2a [112] and shifts in the expression of ATF4 [113] and ATF5 [114]. Such a cascade response provides activation of the protein folding processes. Additionally, this response is facilitated by the induction of HSP60 [115], leading to an increase in collagen synthesis, which is necessary for both cell growth and the formation of intercellular matter. Another factor by which hydrogen may increase its cellular proliferation is its ability to activate GFAP (glial fibrillary acidic protein), a marker of differentiation in glioblastoma cells [116].

An important aspect ensuring the activation of the processes of proliferation, differentiation, and growth of emerging cells is the development of the microenvironment of stem cells. This is facilitated by an increase in the number of colony-forming factors, as well as the regulating effect of cytokines. In particular, the use of H_2_ induces the activation of CCL-2, which leads to a decrease in the level of proinflammatory cytokines (TNFα, IL-6, IFN-y) [113]. The second mechanism determining the decrease in the concentration of these cytokines is the inhibition of NF-kB, which can also be activated by H_2_ [117]. In addition, another antioxidation/detoxification enzyme induced by H_2_ is hemoxygenase-1. This enzyme acts as a strong antioxidant [118] and promotes the stimulation of the synthesis of the anti-inflammatory cytokine IL-10 and the formation of differentiation markers on the cell surface [117]. Although several studies have used various types of cancer cells, the collective results indicate that H_2_ may provide conditions for the accelerated proliferation, differentiation, and growth of stem cells.

This property of H_2_ is of great importance for regenerative medicine since its principal task is to develop the most sparing technologies for stimulating tissue regeneration processes. The combination of molecular, cellular, and tissue effects of H_2_ suggest its multifaceted, pro-regenerative activity, some aspects of which are presented in Figure 7. In particular, the ability of H_2_ to reduce free radical damage is its fundamental value. Relieving pronounced oxidative stress that inevitably occurs in the tissues of a wound or other tissue defect is key for cellular regeneration [119,120]. The creation of favorable conditions for the restoration of the cellular composition of the tissue is also attained by an anti-inflammatory effect [120,121,122] as well as the formation of intercellular substance components (in particular, collagen). Direct replacement of cellular deficiency occurs due to activation of mesenchymal stem cells, and a stimulation of their proliferation and differentiation which occurs under the influence of molecular hydrogen. This process is additionally induced and regulated by a set of pro-regenerative cytokines and cell migration into a definitive niche, which is facilitated by activation of the expression of cell adhesion molecules [123]. In general, the effect of H_2_ on the state of mesenchymal stem cells and on tissue regeneration processes should be considered favorable.

## 6. Conclusions

In conclusion, molecular hydrogen, the lightest and most ubiquitous medical gas, has a wide range of biological activities and is mainly characterized by its antioxidant, anti-inflammatory, and anti-apoptotic actions. It is also involved in the regulation of the expression of numerous genes, the protection of various biomacromolecules from oxidative damage, the stimulation of energy production (ATP) [124,125], and other effects. At the same time, despite the sharp increase in the number of studies and publications on the biomedical applications of molecular hydrogen, the issue of its use as a pro-regenerative agent has not been fully explored. There are numerous advantages in using this molecule due to the wide breadth of molecular responses that it causes. Therefore, conducting targeted research in this area can open new horizons of regenerative medicine and create an innovative technology for accelerated tissue repair.

## Figures and Tables

**Figure 1 antioxidants-12-00636-f001:**
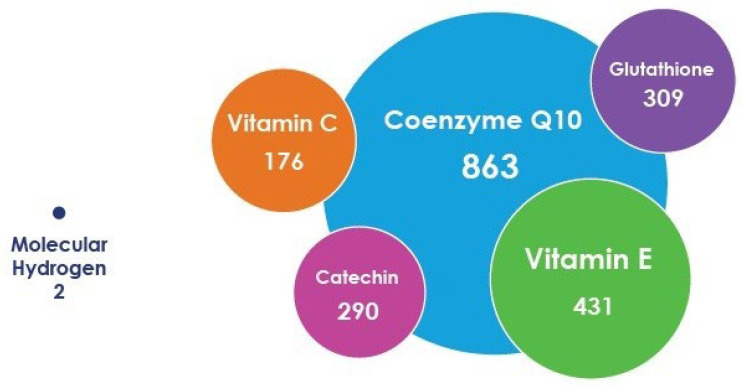
Relative size of molecular hydrogen and some antioxidants based on differences in their molecular weights, not their actual effective diameter size (e.g., kinetic diameter or Stokes–Einstein radius).

**Figure 2 antioxidants-12-00636-f002:**
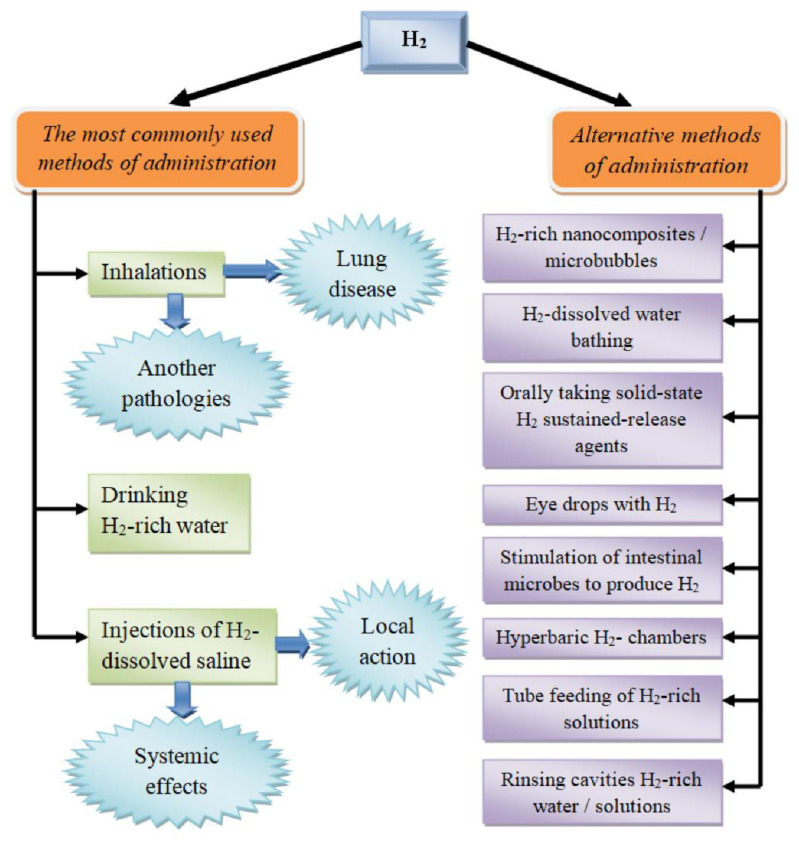
Routes of molecular hydrogen administration.

**Figure 3 antioxidants-12-00636-f003:**
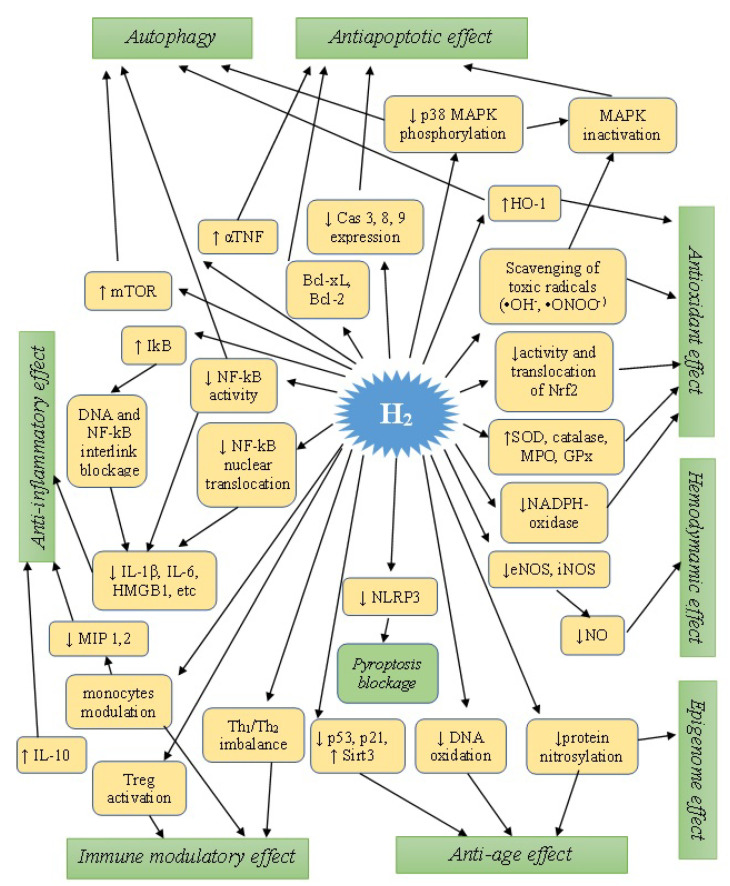
Molecular effects of hydrogen in living organisms. The figure shows some of the proposed mechanisms by which the main effects of hydrogen are mediated on the molecular and cellular level. The brown links represent shifts in regulatory molecules that lead to the development of specific cellular effects (indicated by green blocks). Abbreviations: MAPK–mitogen-activated protein kinase, HO–heme oxygenase, TNF–tumor necrosis factor, SOD–superoxide dismutase, MPO–myeloperoxidase, NOS–nitric oxide synthase (inducible–iNOS; endothelial–eNOS), GPx–glutathione peroxidase, Cas–caspase, HMGB1-high-mobility group protein B1, NLRP–Nucleotide-binding oligomerization domain, Th–T-cytotoxic lymphocyte.

**Figure 4 antioxidants-12-00636-f004:**
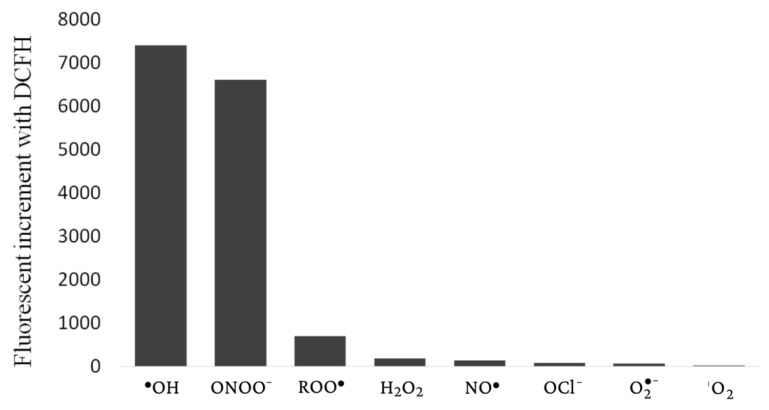
Relative oxidative potential of some bio-oxidants from [73].

**Figure 5 antioxidants-12-00636-f005:**
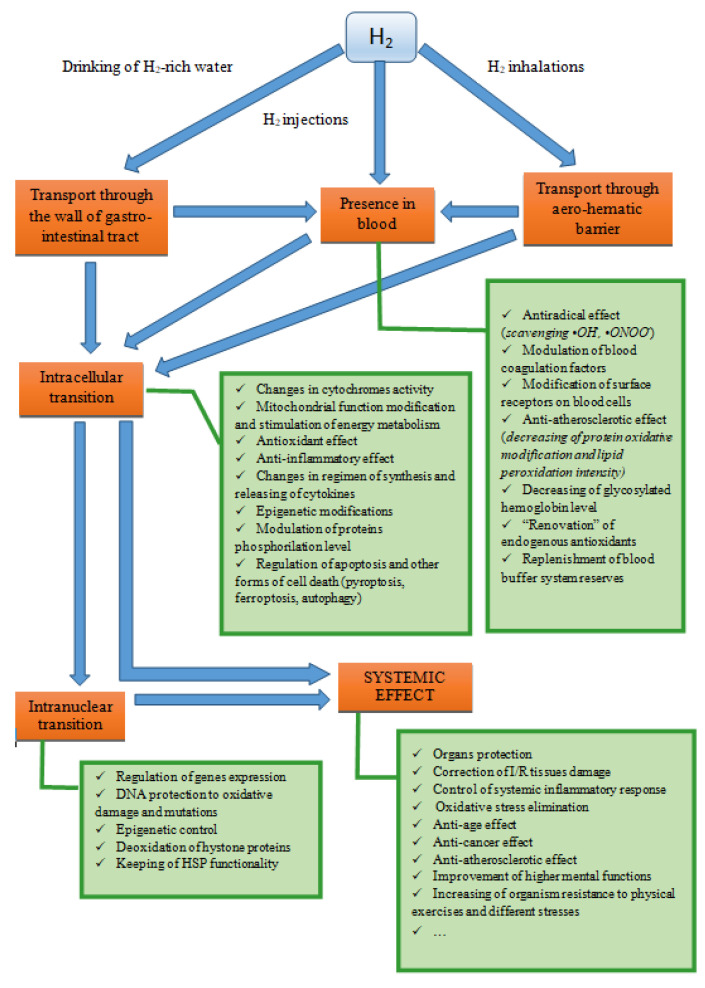
Common cellular, tissue, and systemic effects of molecular hydrogen. The figure demonstrates the pharmacokinetics of molecular hydrogen depending on the route by which it enters the body. In the locations in which hydrogen directly exerts an effect, its molecular mechanisms are shown. Abbreviations: I/R–ischemia and reperfusion, HSP–heat shock protein.

**Figure 6 antioxidants-12-00636-f006:**
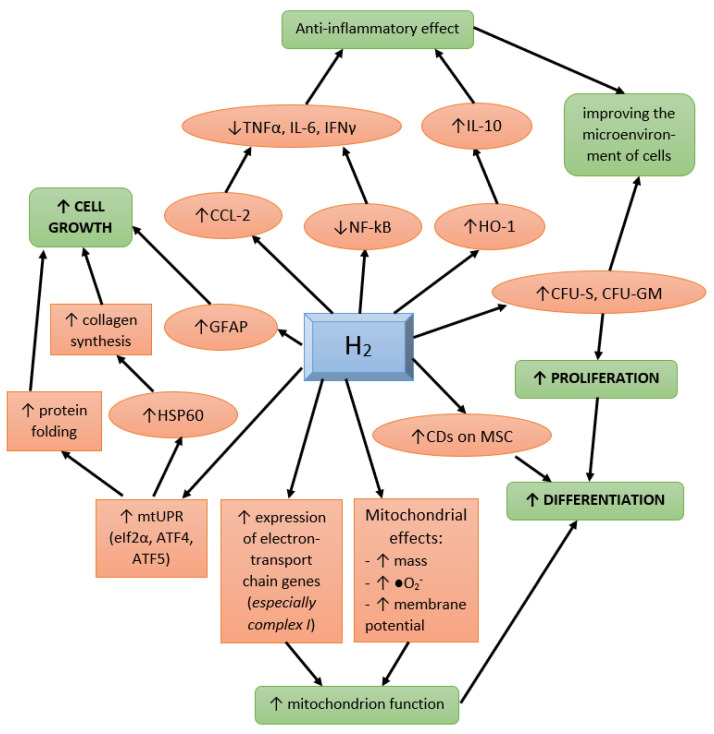
The influence of molecular hydrogen on the state of stem cells. This diagram shows the molecular regulators (indicated by red clouds) and the cellular effects caused by them (green blocks) that are significant for the functioning of stem cells. Abbreviations: HSP–heat shock protein, CFU–colony-forming unit, HO–heme oxygenase, TNF–tumor necrosis factor, IFN–interferon, CD–cluster differentiation marker, MSC–mesenchymal stem cell, GFAP—glial fibrillary acidic protein, IL–interleukin.

**Figure 7 antioxidants-12-00636-f007:**
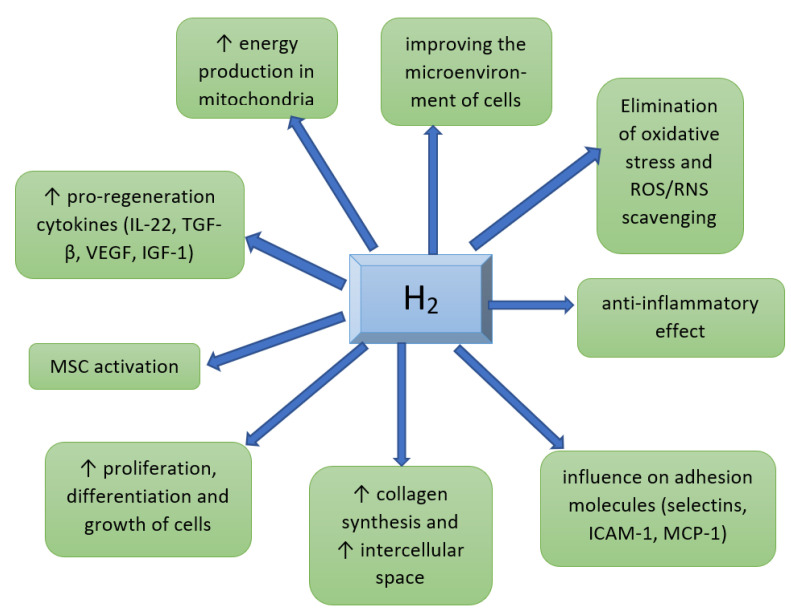
The influence of molecular hydrogen on tissue regeneration. The figure illustrates the effects of molecular hydrogen, potentially significant for stimulating the regeneration and differentiation of stem cells. Abbreviations: IL–interleukin, TGF–tumor growth factor, ROS–reactive oxygen species, RNS–reactive nitrogen species, VEGF–vasculo-endothelial growth factor, IGF—insulin-like growth factor, ICAM-intercellular adhesion molecule, MCP-monocyte chemotactic protein.

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
