# Peer review of "Molecular Hydrogen: From Molecular Effects to Stem Cells Management and Tissue Regeneration"

_antioxidants, 2023, doi:10.3390/antiox12030636_

Round 1

Reviewer 1 Report

The introduction correctly helps to frame and contextualize the work. It presents the current state of the art on several topics which are essential to the review. However, there are some errors that should be corrected. These are some of my comments for improvement:

The abstract contains several repetitions. It must describe briefly in 1-2 small sentences the research method, and then most relevant results. Nevertheless, I believe the abstract should benefit if you add a brief general conclusion.

1.       “Molecular hydrogen” is reported at some times in contracted form “H2, other times in the full form. The format should be consistent.

2.       Line 74. Please correct “hydrogen n oncological” into “hydrogen in oncological”.

3.       Line 75. Review the grammar in the sentence “On the other hand, cellular effects, which serve as the basis for systemic (organismal) and are fundamentally significant for regenerative medicine, are revealed only indirectly”.

4.       Line 88. “in” has been repeated.

5.       Line 369.  “Modulation of autophagy” should be moved new lines in text.

6.       Not all figures are mentioned in the text and are often placed in random places.

7.       Line 440. “This” should be “these”.

Author Response

Reviewer 1

The introduction correctly helps to frame and contextualize the work. It presents the current state of the art on several topics which are essential to the review. However, there are some errors that should be corrected. These are some of my comments for improvement:

The abstract contains several repetitions. It must describe briefly in 1-2 small sentences the research method, and then most relevant results. Nevertheless, I believe the abstract should benefit if you add a brief general conclusion

Dear Reviewer!

Thank you for a thorough analysis of our work and its high appreciation!

All the comments you mentioned were made to the text of the work, the changes are highlighted in blue. Including the abstract, which has been re-written. Below is our point-by-point response to your other comments:

  1. “Molecular hydrogen”is reported at some times in contracted form “H2”, other times in the full form. The format should be consistent.

Response: Thank you for the observation. We tried to be consistent in the abstract and in other places as updated. We attempted to do this throughout the document, but some sentences are suited better to use the full chemical name, whereas other sentences are better to use its chemical symbol. Since, the meaning is not lost, we hope it will be suitable. Additionally, hope the readers will see that these terms are used interchangeably because they are indeed interchangeable, as opposed to an abbreviated form. However, if the editors would like us to make this change, we will do it.

  1. Line 74. Please correct “hydrogen n oncological” into“hydrogen in oncological”.

Response: Thank for the correction, it has been revised.

  1. Line 75. Review the grammar in the sentence “On the other hand, cellular effects, which serve as the basis for systemic (organismal) and are fundamentally significant for regenerative medicine, are revealed only indirectly”.

Response: We revised the sentence as: On the other hand, cellular effects, which are fundamentally significant for regenerative medicine, are revealed only indirectly.

  1. Line 88. “in” has been repeated.

Response: Thank for the correction, it has been revised.

  1. Line 369. “Modulation of autophagy” should be moved new lines in text.

Response: Thank for the correction, it has been revised

  1. Not all figures are mentioned in the text and are often placed in random places.

Response: Thank for the correction, it has been revised

  1. Line 440. “This” should be “these”.

Response: Thank for the correction, it has been revised

Reviewer 2 Report

Review of the paper entitled “Molecular Hydrogen: from Molecular Effects to Stem Cells Management and Tissue Regeneration” by Mikhail Yu. Artamonov, Tyler W. LeBaron, Andrew K. Martusevich, Inessa A. Minenko, Felix A. Piatakovich and Sergey V. Dlin.

      In this paper the Authors summarize some important mechanisms of molecular hydrogen (H2) and its effect on mesenchymal stem cells. For a long time, it was believed that H2 is a biologically inert molecule that does not react with chemical compounds present in the cells of living organisms. Interest in the physiological and therapeutic role of H2 began in 2007 when Ohsawa et al. showed that the inhalation of H2 gas markedly suppressed brain injury by buffering the effects of oxidative stress [Ohsawa I, Ishikawa M, Takahashi K, Watanabe M, Nishimaki K, Yamagata K, Katsura K, Katayama Y, Asoh S, Ohta S. Hydrogen acts as a therapeutic antioxidant by selectively reducing cytotoxic oxygen radicals. Nat Med. 2007 Jun;13(6):688-94. doi: 10.1038/nm1577. Epub 2007 May 7. PMID: 17486089]. Currently, the therapeutic potential of molecular hydrogen has been widely evaluated in various diseases.

The paper presented by the Authors is a very interesting and comprehensive review.

 My comment

     In order to increase the quality and attractiveness of this manuscript, the papers on endogenous sources of molecular hydrogen should be cited and discussed (for example Jahng J, Jung IS, Choi EJ, Conklin JL, Park H. The effects of methane and hydrogen gases produced by enteric bacteria on ileal motility and colonic transit time. Neurogastroenterol Motil. 2012 Feb;24(2):185-90, e92. doi: 10.1111/j.1365-2982.2011.01819.x. Epub 2011 Nov 20. PMID: 22097886), as well as those papers that concern the impact of molecular hydrogen on the metabolic pathways of carbohydrates, fats and amino acids [for example Adzavon YM, Xie F, Yi Y, Jiang X, Zhang X, He J, Zhao P, Liu M, Ma S, Ma X. Long-term and daily use of molecular hydrogen induces reprogramming of liver metabolism in rats by modulating NADP/NADPH redox pathways. Sci Rep. 2022 Mar 10;12(1):3904. doi: 10.1038/s41598-022-07710-6. PMID: 35273249; PMCID: PMC8913832].

Author Response

Reviewer 2

Review of the paper entitled “Molecular Hydrogen: from Molecular Effects to Stem Cells Management and Tissue Regeneration” by Mikhail Yu. Artamonov, Tyler W. LeBaron, Andrew K. Martusevich, Inessa A. Minenko, Felix A. Piatakovich and Sergey V. Dlin.

      In this paper the Authors summarize some important mechanisms of molecular hydrogen (H2) and its effect on mesenchymal stem cells. For a long time, it was believed that His a biologically inert molecule that does not react with chemical compounds present in the cells of living organisms. Interest in the physiological and therapeutic role of Hbegan in 2007 when Ohsawa et al. showed that the inhalation of H2 gas markedly suppressed brain injury by buffering the effects of oxidative stress [Ohsawa I, Ishikawa M, Takahashi K, Watanabe M, Nishimaki K, Yamagata K, Katsura K, Katayama Y, Asoh S, Ohta S. Hydrogen acts as a therapeutic antioxidant by selectively reducing cytotoxic oxygen radicals. Nat Med. 2007 Jun;13(6):688-94. doi: 10.1038/nm1577. Epub 2007 May 7. PMID: 17486089]. Currently, the therapeutic potential of molecular hydrogen has been widely evaluated in various diseases.

The paper presented by the Authors is a very interesting and comprehensive review.

 My comment

     In order to increase the quality and attractiveness of this manuscript, the papers on endogenous sources of molecular hydrogen should be cited and discussed (for example Jahng J, Jung IS, Choi EJ, Conklin JL, Park H. The effects of methane and hydrogen gases produced by enteric bacteria on ileal motility and colonic transit time. Neurogastroenterol Motil. 2012 Feb;24(2):185-90, e92. doi: 10.1111/j.1365-2982.2011.01819.x. Epub 2011 Nov 20. PMID: 22097886), as well as those papers that concern the impact of molecular hydrogen on the metabolic pathways of carbohydrates, fats and amino acids [for example Adzavon YM, Xie F, Yi Y, Jiang X, Zhang X, He J, Zhao P, Liu M, Ma S, Ma X. Long-term and daily use of molecular hydrogen induces reprogramming of liver metabolism in rats by modulating NADP/NADPH redox pathways. Sci Rep. 2022 Mar 10;12(1):3904. doi: 10.1038/s41598-022-07710-6. PMID: 35273249; PMCID: PMC8913832].

Response

Thank you for a thorough analysis of our work and its high appreciation!

All the comments you mentioned were made to the text of the work, the changes are highlighted in blue. Below is our point-by-point response

  1. The issues of endogenous generation of molecular hydrogen by intestinal microflora are discussed. The relationship between the hydrogen-producing activity of the microbiota and its composition is presented. It is shown that the generation of hydrogen depends on the ratio of microorganisms-producers and consumers.
  2. A separate emphasis is placed on the metabolic effect of molecular hydrogen, including its effect on various components of metabolism.
  3. We have included the suggested references as highlighted in the manuscript.

Reviewer 3 Report

This is an interesting and rather well-written review on mainly beneficial roles of H2. However, there are also shortcomings such as several statements requiring appropriate references and some misinterpretations (for example on the role of inflammation and ROS in wound healing where references are missing throughout the entire paragraph) as well as some overgeneralisation of results and studies performed on cancer cells yet the authors prescribe them to stem cells. 

I list major points below:

1. line 37: Please explain "sanogenetic" as I am sure that many readers are not aware of its meaning.

2. line 46: In which category (medicine, Physiology, chemistry etc) was the Nobel prize given?
3. line 86: You mention but do not describe how H2 is generated endogenously.

4. Line 121: Please give the reference "Doyle et al., 1975 a number as all other references and do not give initials, just last names.

5. The statement in line 125 requires a proper reference, just citing your own figure is not sufficient.

6. line 165: I am wondering whether instead of "bioavailability" you rather mean "uptake" as in my view "bioavailability" rather relates to what fraction really reaches its target location.

7. On age 6 you rather explain transiency after H2 uptake-so what are the approximate times as in line 228 you just say longer or shorter without giving any idea to the reader not from the field whether we talk about seconds, minutes, hours or even longer.

8. Please give better and more detailed figure legends, at least starting from figure 3. Please also explain all abbreviations from the figures as well those used in the text, for example UPR, DGFH (in fig. 4 y-axis label while in the text you say that the paper used DCF) etc.

9. line 331: what do you mean with "apoptosis... can ENTER various...processes"? Perhaps rephrase and make clearer.

10. line 382: Please provide a reference for the statement and also better explain what you mean with "processes of consideration".

11. The sentence in lines422/3 is not clear to me. What do you mean with "presupposes their (who's?) implementation through the management of the state of cells"? Please clarify and rephrase.

12. Line 430: There is not such a thing as £definitive cells". Do you rather mean "differentiated"?

13. The statement in line 431 requires a reference. "It is known" is not sufficient, evidence required.

14. Reference 107 is about cancer cells and various different types (responders vs non-responders), thus I don't think you can generalise the findings of this study to other, normal cell types. For example, increase of mitochondrial mass and ROS are characteristics of senescent normal somatic cells and are associated with a stop of proliferation and thus the opposite of what cancer cells do. Thus, please put your statements in the correct context of cells.

15. line 441/2: What exactly do you mean with "mtUPR as a mediator of mitochondrial origin"? Please clarify and rephrase.

16. lines 443/4: What exactly has the unfolded protein response to do with a translation initiation factor? Please clarify and give more detail.

17. line 448: GFAP (give abbreviation) is specific for glia cells in the brain and the paper is about glioblastoma, thus another cancer cell type. Please emphasise this and put into context of the specific cell types and do not overgeneralise such specific findings.

18. line 451: What is "coordinated evolution" in the context of microenvironment of stem cells? I don't think "evolution" is the correct term here-perhaps use "development".

19. 462/63: You conclusion is about stem cells, meaning somatic stem cells, not cancer cells or cancer stem cells, thus please doublecheck that all your provided statements are really valid for stem cells and not obtained in other, mainly cancer cells!

20. lines 466-482: You do not provide a single reference for your statements in the entire paragraph! Wound healing requires ROS, inflammation and even senescence, thus inhibiting these processes does not improve but hinder wound healing. There is quite a lot of literature on this, for example Khalid et al., 2022.

21. Figure 7: I don't really see any evidence for changes/increases in energy generation in mitochondria, as just an increase in gene expression for single genes, for example, of complex I are not any evidence. Please provide real data on more ATP under H2 changes and best give references for all these  different effects.

spelling and grammar mistakes:
line 74: Please add an "i" for "in oncological diseases", line 88: Remove one redundant "in". line 113: I would recommend to replace "wide" with "broad" and in line 119: replace "is" with "was" as it is past tense. line 132: better to say "A rat model". line 245: Please remove "Thus" at the start of this heading and sentence as it is not obvious what it refers to. line 329: Please replace "method" with "pathway", line 336: better say "It has been shown" as it is past tense. line 356: use "process" instead of plural for apoptosis which is just one. line 369: Move heading to new line as it starts a new paragraph. line 393: Please use "inhalation" just in singular. line 394: better use "location" instead of "localisation". line 400: please replace "entrance" with something more suitable like "entering". line 408: Please replace "renovation" as it is not a flat you renovate. line 443"THE stimulating effect..." line 494: better say "wide range"

Author Response

Reviewer 3

This is an interesting and rather well-written review on mainly beneficial roles of H2. However, there are also shortcomings such as several statements requiring appropriate references and some misinterpretations (for example on the role of inflammation and ROS in wound healing where references are missing throughout the entire paragraph) as well as some overgeneralisation of results and studies performed on cancer cells yet the authors prescribe them to stem cells. I list major points below:

Dear Reviewer!

Thank you for a thorough analysis of our work and its high appreciation!

Taking into account your comments, the text of the article has been significantly revised. All the comments you mentioned were made to the text of the work, the changes are highlighted in blue. Below is our point-by-point response

The following changes have been made to the text:

  1. The term "sanogenetic" has been replaced by "beneficial".
  2. The category in which the Nobel Prize was awarded is indicated.
  3. The mechanisms of hydrogen generation by intestinal microflora are described.
  4. The link to Doyle's research is given, the mention of the surnames of the authors of the articles in the text is corrected.
  5. A reference was added to the statement with the figure
  6. Re bioavailability and uptake, we revised it to say uptake per your suggestion
  7. The time intervals of the effect of hydrogen at different routes of administration have been clarified.
  8. Detailed explanations of the figures are provided, all abbreviations used in them are deciphered.
  9. We rephrased our statement on apoptosis to clarify the sentence.
  10. Rephrased the process of consideration to refer to autophagy and added citation.
  11. the “presupposes” phrase was revised for clarity.
  12. Thank you for noting our error, “definitive cells” was changed to “differentiated cells”.
  13. We removed the sentence with “It is known” because it was not necessary for our discussion and lacked clarity.
  14. We revised the paragraph about the influence of H2 on the mitochondria to specify that the reference was regarding cancer cells.
  15. The phrase of mtUPR as a mediator of mitochondrial origin was removed and the section revised.
  16. The role of mtUPR on the effect on translation initiation has been clarified.
  17. The participation of GFAP in the hydrogen effect only in glial cells has been clarified, and we specified that this was in glioblastoma.
  18. We agree and changed coordinated evolution to development per your suggestion.
  19. We rephrased our statement so that it was not conflating cancer cells with stem cells, but only indicative of its potential actions.
  20. References are provided confirming the importance of limiting oxidative stress and inflammation in creating optimal conditions for regeneration.
  21. We added references 125 and 126 to help substantiate the idea that H2 increases energy (ATP) production in addition to what we already discussed regarding the effects on the mitochondria.

Spelling and grammar mistakes

All grammatical and punctuation errors indicated by you have been corrected.

Round 2

Reviewer 3 Report

The authors addressed the majority of my comments and greatly improved the review quality. There are just very few minor issues that still need to be considered.

1. line 339: Apoptosis is not a method, but a process. Methods are exclusively used by humans.

2. line 341: it should be "processES" as plural is required.

3. line 415. Please replace "localizations" with "locations" 

4. Please remove the term "renovation" in the context of antioxidants and replace with something more suitable like "renewal, activation etc"

5. When you talk about proliferative activity you have to add of what? I suppose you refer to cancer cells, not mitochondria which in the way it stands currently is not unambiguous.

Author Response

Reviewer comment

The authors addressed the majority of my comments and greatly improved the review quality. There are just very few minor issues that still need to be considered.

Response: Thank you for your assistance in helping us improve the manuscript.

  1. line 339: Apoptosis is not a method, but a process. Methods are exclusively used by humans.

You are right. Thank you for noticing this error. We fixed it.

  1. line 341: it should be "processES" as plural is required.

We fixed it, this should be plural. Thank you again.

  1. line 415. Please replace "localizations" with "locations" 

We also fixed this error. Thank you.

  1. Please remove the term "renovation" in the context of antioxidants and replace with something more suitable like "renewal, activation etc"

Thank you. We changed it to “the rejuvenation of endogenous antioxidants”

  1. When you talk about proliferative activity you have to add of what? I suppose you refer to cancer cells, not mitochondria which in the way it stands currently is not unambiguous.

Thank you for noticing this. We changed it as follows: “…an increase in cellular proliferation, at least in some types of cancer cells”---(which is the study/cell what we were just talking about)